# AI-Driven Refactoring: A Pipeline for Identifying and Correcting Data Clumps in Git Repositories

**Nils Baumgartner** [1,*] **, Padma Iyenghar** [2,3] **, Timo Schoemaker** [1] **and Elke Pulvermüller** [1]

1   Software Engineering Research Group, School of Mathematics, Computer Science, Physics, Osnabrück University, 49090 Osnabrück, Germany; tschoemaker@uni-osnabrueck.de (T.S.); elke.pulvermueller@uni-osnabrueck.de (E.P.)
2   Faculty of Engineering and Computer Science, University of Applied Sciences Osnabrück, 49090 Osnabrück, Germany; piyengha@uni-osnabrueck.de
3   Innotec GmbH, Hornbergstrasse 45, 70794 Filderstadt, Germany
*   Correspondence: nils.baumgartner@uni-osnabrueck.de; Tel.: +49-541-969-2695

**Abstract:** Data clumps, groups of variables that repeatedly appear together across different parts of a software system, are indicative of poor code structure and can lead to potential issues such as maintenance challenges, testing complexity, and scalability concerns, among others. Addressing this, our study introduces an innovative AI-driven pipeline specifically designed for the refactoring of data clumps in software repositories. This pipeline leverages the capabilities of Large Language Models (LLM), such as ChatGPT, to automate the detection and resolution of data clumps, thereby enhancing code quality and maintainability. In developing this pipeline, we have taken into consideration the new European Union (EU)-Artificial Intelligence (AI) Act, ensuring that our pipeline complies with the latest regulatory requirements and ethical standards for use of AI in software development by outsourcing decisions to a human in the loop. Preliminary experiments utilizing ChatGPT were conducted to validate the effectiveness and efficiency of our approach. These tests demonstrate promising results in identifying and refactoring data clumps, but also the challenges using LLMs.

**Keywords:** data clumps; AI-driven refactoring; software quality; large language models; continuous integration and delivery; EU AI act; code smells; software maintenance; ChatGPT; code improvement

## 1. Introduction

In the current emerging wave of innovations in AI, the landscape of technology is evolving at an unprecedented pace. This surge in development serves as a catalyst for leading-edge technologies and greatly accelerates digital transformation initiatives. Consequently, there is a notable surge in interest and investments within this sector. For instance, a McKinsey global survey report (https://www.mckinsey.com/featured-insights/mckinsey-global-surveys, accessed on 18 March 2024) on AI makes a bold prediction that AI could contribute up to USD 13 trillion to global economic activity by 2030. This represents about 16% higher cumulative GDP compared to today. Moreover, the above analysis also identified software engineering as one among the 4 major fields that could account for approximately 75% of the total annual value from AI-based use cases. Thus, the ascent of AI represents a transformative force within the software industry, with the potential to automate a substantial portion of tasks traditionally performed by humans [1].

In the field of software engineering, Continuous Integration and Delivery (CI/CD) methodologies have emerged as indispensable tools for enhancing the quality of software products. They contribute to bolstering reliability, maintainability, and overall development processes. Notably, CI/CD practices facilitate the shortening of release cycles, thereby fostering agility and efficiency within software development endeavours [2]. As software projects expand in scale and complexity, they inevitably encounter additional challenges and complexities. Issues such as code smells, architectural bottlenecks, and scalability

concerns become more pronounced [3]. Addressing these challenges requires a combination of robust development practices, effective project management, and strategic utilization of technological solutions. Furthermore, the integration of DevOps practices brings about a transformation in agile methodologies, extending their applicability from small teams to encompass extensive, large-scale applications. This evolution necessitates the establishment of complex organizational structures capable of adapting to the dynamic requirements of software development [4].

Enhancing software quality emerges as a crucial strategy to mitigate high maintenance costs, which often constitute a significant portion, ranging from 40% to 80% of total expenses This investment yields substantial dividends within a relatively short timeframe, often becoming evident within a few weeks [5,6]. Refactoring [7] stands out as a fundamental technique for enhancing software quality. It involves restructuring code to improve its internal structure and organization while preserving its observable functionality. By systematically refactoring codebases, developers can eliminate technical debt, enhance maintainability, and pave the way for more efficient development practices. Integrating AI-driven innovations with software quality improvement through techniques like refactoring presents a promising avenue for optimizing software development processes and outcomes in the evolving landscape of technology [5,6].

As software development methodologies evolve to embrace automation and efficiency-enhancing practices such as CI/CD and DevOps, there is a growing recognition of the importance of addressing inherent code and design smells to maintain and enhance software quality. One such prevalent issue is the presence of data clumps, which signify redundancy and inefficiency within software projects. Data clumps are groups of variables that appear repeatedly in different places in a software project. Currently, these data clumps cannot be refactored automatically. Data clumps are a manifestation of code smells in source code [7] and design smells in, for example, class diagrams [8]. Recognizing the significance of addressing data clumps in improving software quality, the next logical step is to explore automated solutions to refactor these clumps, leveraging advancements in AI [9].

On the other hand, to maintain trust and safety in software development leveraging AI, careful consideration of its integration is paramount. Despite its significant potential, unchecked integration can introduce substantial risks. The rapid evolution and widespread availability of AI methodologies, including LLMs, underscore the importance of mitigating these risks to ensure the maturity of AI-driven software systems. Regulatory measures, such as the recently introduced EU AI Act [10], play a crucial role in this regard by imposing stringent requirements, including risk management and data governance, tailored to various AI applications and purposes. When developing AI-driven refactoring pipelines, adherence to such regulations is imperative to foster the creation of dependable and safe AI applications, paving the way for innovative yet compliant automatic refactoring developments [10].

However, the use of AI poses several challenges that need to be overcome. Many AI models are so complex that they are beyond our understanding. To enable trust in autonomous software, the necessity of human agency and oversight is emphasized in the EU-ethical guidelines for trustworthy AI [10,11]. This means that a human should have the overview, final decision, and responsibility for the actions of an AI system [11,12].

The objective of this paper is to outline and construct a pipeline tailored for the automated detection and refactoring of data clumps within publicly available Git projects. The proposed implementation envisages to integrate the human-in-the-loop methodology, leveraging human decisions to refine rejected refactorings in subsequent iterations. This iterative approach aims to tackle the challenges posed by the complexity of AI models and ensure the trustworthiness of the refactoring process. By combining automated techniques with human oversight and intervention, the pipeline endeavours to enhance the efficiency and effectiveness of data clump refactoring, thereby contributing to the advancement of software quality and maintainability. We expect our work to be particularly helpful in large projects, as in these projects, the overview by humans over the entire project decreases,

thereby leading to potential errors or quality deterioration. By combining deterministic detection and refactoring, as well as the linguistic processing and understanding of the context by AI, an enrichment of a virtual assistant is expected. Based on the aforesaid objectives, the novelties in this paper include the following:

- Introduction of an AI-driven tailored pipeline for data clump refactoring: this pipeline leverages the capabilities of LLMs such as ChatGPT, to automate the detection and resolution of data clumps, thereby enhancing code quality and maintainability.
- Integration of human-in-the-loop methodology: by incorporating human decisions to refine rejected refactorings in subsequent iterations, the pipeline aims to enhance the effectiveness and trustworthiness of the refactoring process.
- Consideration of regulatory standards such as the EU-AI Act: this work carefully considers the latest EU-AI Act ensuring compliance with the latest regulatory requirements and ethical standards for the use of AI in software development.
- Preliminary experimentation and validation: these tests demonstrate promising results in identifying and refactoring data clumps, while also highlighting the challenges associated with using LLMs.

Following this introduction section, the rest of this paper is organized as follows. Section 2 provides an overview of related works and a literature review. Subsequently, our methodology is described in more detail in Section 3. Section 4 deals with an insight into our implementation. The resulting experiments and results are presented in Section 5. A discussion of these results then takes place in Section 6. Finally, the conclusion and future work are provided in Section 7.

## 2. Literature Review

In this section, the literature review is organized into several distinct groups, each focusing on a specific aspect of software engineering and refactoring namely, software engineering pipelines, LLMs in Software Engineering, AI-Driven refactoring, and the identification and resolution of data clumps and code smells in software refactoring. Each group provides a comprehensive overview of relevant research, supplemented by brief background information where appropriate. This section concludes with a summary of the existing gaps in the current state of research.

### 2.1. Software Engineering Pipelines

Software engineering pipelines have significantly evolved over the past years. While traditional manual processes in the 1980s took several years to develop and deploy new features, today, several hundred deployments per day are not uncommon. Modern pipelines can be highly integrated and automated, comprising complex systems themselves, and enable continuous integration and continuous deployment [13].

The adoption and use of CI/CD in the software development process also introduce common challenges. The primary challenges include insufficient team awareness and communication, which prevent the transition to continuous practices. Implementing CI/CD often requires additional costs, resources, and training, necessitating new technical and soft skills. Organizations also frequently face resistance due to scepticism, distrust, and the fear of losing jobs, which hinders the transition [14].

A critical factor for successful CI/CD is transparency for all team members regarding the number of errors, the status of the project, who broke the build, and the time when features are completed. Long-running tests, manual tests, and a high number of faulty tests are factors that contribute to unsuccessful CI/CD implementations [14]. For CI/CD pipelines to be effective and useful, they require ongoing maintenance and configuration, for example, when system requirements change [15].

There are a variety of pipeline frameworks, which often interconnect. Some of the well-known frameworks include Jenkins (https://www.jenkins.io/, accessed on 18 March 2024), GitLab (https://docs.gitlab.com/ee/ci/, accessed on 18 March 2024), and TeamCity

(https://www.jetbrains.com/teamcity, accessed on 18 March 2024). These frameworks can also be interconnected in various ways.

### 2.2. LLMs in Software Engineering

In the landscape of AI, in recent years deep learning [16] and LLMs has shifted the technology from a niece speciality to a near-ubiquitous tool, finding applications across diverse domains ranging from image generation to textual analysis and language processing [12]. A Large Language Model is a very large deep learning model that is pre-trained on a massive amount of data. Deep learning is a form of Machine Learning (ML), which is also a neural network, but with additional layers. LLMs are known for their ability to be trained on domain-specific datasets or across a broad spectrum of general knowledge, making them highly versatile.

A well-known LLM is OpenAI's Generative Pre-trained Transformer series (GPT) with GPT-3 and GPT-4 [17]. These offer a powerful chatbot interface capable of communicating in normal, human-like English and able to complete a wide range of tasks. The integration of LLMs into software engineering extends beyond theoretical applications. For instance, the paradigm of Model-Based Software Engineering (MBSE) is enhanced by LLMs, by offering even a machine-executable version of the traditional V-Model [18], enriching the design and development process significantly.

The ranges of the applications of LLMs in software engineering can be further divided as listed below. However, it is important to acknowledge that the field of LLM applications in software engineering is continually evolving, with new use cases emerging over time as technology advances.

- Chatbots in Software Engineering: Chatbots, powered by LLMs, are redefining interactions within software development environments and offering real-time assistance, for instance, for system architects [19].
- Uses in Teaching, Commentary, and Requirement Analysis: Beyond coding, LLMs find also application in the education settings for teaching programming concepts, generating code commentary for better understanding, and automated requirement analysis processes to help developers to ensure accurate software specifications [20].
- Code Generation, Documentation, and Debugging: The use of AI models can help to increase the effectiveness during programming and can help to detect bugs, aiding in the debugging and also the generation of automatic test cases [21].
- Natural Language Generation and Understanding: Leveraging LLMs, such as GPT models, allows for the interpretation and processing of natural language inputs within software development contexts. These models can generate code documentation, comments, or even entire program structures based on textual descriptions, thereby streamlining the development process and improving collaboration among team members [22].
- Automated Code Refactoring and Optimization: By leveraging LLMs, software systems can automate the process of refactoring and optimizing code to enhance performance, readability, and maintainability. These models excel in identifying redundant code, optimizing algorithms, and suggesting structural improvements, thereby reducing technical debt and bolstering the long-term viability of software projects [23]. However, it is worth noting that existing research in this area may not fully consider the unique challenges posed by large-scale Git projects and regulatory requirements such as the EU-AI Act and other relevant regulations. Hence, there is a need to explore how these models can effectively navigate such complexities while adhering to regulatory standards. This paper specifically focuses on an AI-driven refactoring pipeline for identifying and correcting data clumps in Git repositories.
- Predictive Maintenance: Utilizing LLMs for predictive maintenance involves analysing historical data and system behaviours to forecast potential software failures before they occur. By leveraging machine learning algorithms, these models can identify patterns indicative of impending issues, enabling proactive maintenance interventions to minimize downtime and ensure system reliability [24].

- Automated Code Review and Quality Assurance: LLMs can automate code review processes by analysing codebases against predefined coding standards, identifying potential bugs, style violations, or security vulnerabilities. Integrating LLM-powered code review tools into development pipelines streamlines the review process, accelerates feedback cycles, and improves overall software quality [25].
- Semantic Code Search: Enhanced by LLMs, semantic code search systems can understand the context and intent behind code queries, enabling more accurate and relevant search results. By considering factors such as variable names, function signatures, and comments, these systems can assist developers in finding code snippets that match their specific requirements, increasing productivity and code reuse [26].
- Automated Software Testing: Leveraging LLMs for automated software testing involves generating test cases, inputs, or assertions based on code specifications and requirements. These models can analyse code semantics and behaviour to automatically generate comprehensive test suites, covering various execution paths and edge cases, thereby improving test coverage and the reliability of software systems [27].
- Automated Project Management and Scheduling: Integrating LLMs into project management systems enables automation of tasks such as task allocation, scheduling, and resource management. These models can analyse project requirements, team capacities, and external factors to optimize project timelines, allocate resources efficiently, and mitigate risks, facilitating smoother project execution and delivery [28]
- Automated Code Summarization and Abstraction: LLMs can automatically summarize code functionality and abstract complex code segments into more manageable representations. By analysing code semantics and execution paths, these models can generate concise summaries, diagrams, or higher-level abstractions, aiding developers in understanding and maintaining large codebases effectively [29,30].

Thus, it is clear that continuing to transform various facets of software engineering, LLMs are already widely used and versatile tools in the software development process, revolutionizing various aspects. From redefining interactions through chatbots to facilitating education and requirement analysis, LLMs offer a wide range of applications. They enhance code generation, documentation, and debugging, while also enabling natural language understanding and processing. Additionally, LLMs play a crucial role in automated code refactoring and optimization, predictive maintenance, and code review. They facilitate semantic code search, automate software testing, and streamline project management tasks. Furthermore, LLMs aid in automating code summarization and abstraction, contributing to improved software quality and maintainability.

Listing 1 displays a simple example on how a LLM could be queried to create a python calculator. The differences between various services or LLMs are often minimal. Therefore, tokens are frequently required for access to online LLM services due to payment obligations. This simple example demonstrates that a clear prompt is often crucial to receive the exact desired result. This necessity gave rise to the term prompt engineering [31,32].

**Listing 1.** API request example.

```
1 curl http://localhost:11434/api/chat -d '{
2   "model": "codellama",
3   "messages": [
4     {
5       "role": "developer",
6       "content": "Write me a python calculator"
7     }
8   ],
9   "stream": false
10 }'
```

While there are clear advantages to using LLMs, it is important to also acknowledge their limitations and challenges such as the ones listed below (but not limited to), namely:

- Despite producing coherent and seemingly accurate responses, LLMs lack true sentience. This can lead to outputs that are logical and confident yet incorrect, which, in critical real-world scenarios, could have significant consequences. Thus, while the benefits of LLMs can appear promising, there is always a risk of inaccuracies, necessitating subsequent checks to ensure the reliability and appropriateness of the response, especially in sensitive situations.
- Another drawback is that LLMs often incorporate randomness in their responses, meaning the same inquiry may yield different and also sometimes incorrect results.
- Additionally, some models may not adhere to the desired output format, requiring further adjustments.
- LLMs are typically black boxes, making it challenging to understand the processes behind their responses. These models do not inherently access the most up-to-date information but require continual retraining with new data or capabilities to access the latest information.
- Regarding security concerns, particularly when generating code, it is essential to recognize that the output may not be secure or reliable, demanding additional verification methods.

These limitations of the present LLMs do affect an AI-driven software development. Such affects are that large software projects cannot be passed at one into a LLM due to the limited context window, although there are approaches with splitting the content into smaller manageable parts or using a vectored search, additional steps are required. Other limitations such as incorrect formats and final testing and acceptance will still need a human in the loop. This challenges of limitations also come into play when considering refactoring large software projects.

### 2.3. AI-Driven Refactoring

Code refactoring is a critical process in software development, aiming to improve the software quality without changing its external behaviour. This helps to maintain code readability, reducing complexity, thus helping developers to further implement new features and reducing the onboarding time. A clean and efficient codebase facilitates easier updated, debugging, and comprehension by various members of the development team.

The usage of AI in software engineering has improved the traditional practices and also code refactoring being no exception. AI-driven refactoring tools are using machine learning algorithms and LLMs to analyse code patterns, identifying refactoring opportunities, and implement code changes autonomous [33]. This usage of AI extends a basic suggestion systems to include complex refactorings, thereby reducing manual effort and minimizing human error. Despite reducing the human error, we are now facing the challenge to handle errors made by AI. Given this background, the following lists some of the critical challenges in the usage of AI-driven refactoring.

- Regulatory Compliance and EU AI Act: Adhering to local laws and regulations, such as the EU AI Act [10], presents a significant challenge for AI-driven refactoring. Developers must ensure that their tools and practices comply with legal requirements regarding AI usage, data privacy, and ethics. Solutions involve staying updated on relevant regulations, conducting legal reviews, and implementing measures to ensure compliance with regulatory standards.
- Accuracy and Reliability: Ensuring the accuracy and reliability of AI-driven refactoring tools is crucial. AI models may sometimes produce inaccurate or unreliable refactorings, leading to unintended consequences or errors in the codebase. Solutions involve rigorous testing, validation, and continuous improvement of AI models to enhance their accuracy and reliability over time.

- Interpretability and Explainability: Providing explanations or rationales for each refactoring suggestion helps developers understand why certain refactorings are recommended and how they will impact the codebase. Solutions involve designing AI models and tools with built-in explainability features to improve interpretability and explainability.
- Integration with Existing Workflows: Integrating AI-driven refactoring tools into existing development workflows can be challenging. Solutions include providing easy-to-use APIs, plugins, or integrations that developers can use with their preferred IDEs or version control systems.
- Scalability and Performance: AI-driven refactoring tools must be scalable and performant to handle large codebases efficiently. Solutions involve optimizing algorithms and infrastructure to handle large-scale codebases effectively and efficiently.
- Privacy and Data Security: AI-driven refactoring tools often require access to sensitive code repositories and data, raising concerns about privacy and data security. Solutions include implementing robust security measures, such as encryption, access controls, and data anonymization, to protect sensitive information and prevent unauthorized access or misuse.
- Bias and Fairness: AI models used in refactoring tools may exhibit bias or unfairness, leading to unequal treatment or outcomes for certain groups of developers or codebases. Solutions involve conducting bias assessments, implementing fairness-aware algorithms, and ensuring diverse and representative training data to mitigate bias and promote fairness in AI-driven refactoring.
- Ethical Considerations: AI-driven refactoring introduces ethical considerations related to the potential impact on developers, code quality, and the broader software development ecosystem. Ethical challenges may include issues such as fairness, transparency, accountability, and unintended consequences. Solutions involve incorporating ethical guidelines and principles into the design and deployment of AI-driven refactoring tools, fostering discussions within the software development community, and ensuring that AI systems prioritize ethical values and considerations.

Regulatory Compliance and EU AI Act

Challenges and solution in the usage of AI has to comply by local laws, for instance where different purposes and applications underline specific laws when using AI [10]. The EU commission design a risk-based approach of different risk statuses, which result in different prohibitions. Since an AI-driven refactoring approach could be applicable to any software, the usage for specific projects cannot be applied. The different risk statuses are unacceptable risk, high risk, limited risk, and low or minimal risk.

- Unacceptable risk: Prohibited AI practices: AI practices that are considered to be a clear threat to people's safety, livelihood, and right are banned. Therefore AI-Driven Refactoring cannot be used here.
- High risk: Regulated high-risk AI systems: AI Systems or which use AI generated code fall in this category when either: products falling under the EU's product safety legislation (including toys, aviation's, card, medical devices, and lifts) or systems falling into eight specific areas (like biometric identification, education and vocational training, employment, law enforcement, access to enjoyment of essential private services, and management and operation of critical infrastructure). Systems with this status have to comply with a range of requirements such as high quality of the datasets, risk assessment, logging of activity, detailed documentation, clear and adequate information to the user, high level of robustness, testing and human oversight.
- Limited risk: AI systems such as chatbots with limited risk are obligated with transparency, so that users are aware that they are interacting with a machine.
- Minimal or no risk: Applications such as AI-enabled video games or spam filters fall under this category, which allows the free use of them.

Thus, we can understand that under the EU AI Act, certain AI applications are categorized based on their risk level, with different requirements and prohibitions applying to each category. For example, high-risk AI systems, such as those used in critical infrastructure or healthcare, are subject to stringent requirements, including risk assessment, logging of activity, detailed documentation, transparency, and human oversight. Developers of AI-driven refactoring tools must determine the risk level of their systems and ensure compliance with the corresponding regulatory requirements.

To address these challenges, developers must stay updated on relevant regulations and guidelines, such as the EU AI Act, and conduct legal reviews to assess the compliance of their tools and practices. This may involve consulting legal experts with expertise in AI regulation and data privacy laws to ensure that their tools and practices meet regulatory standards. Implementing measures to ensure compliance with regulatory standards is essential for developers of AI-driven refactoring tools. This may include incorporating privacy-preserving technologies, such as differential privacy or federated learning, to protect sensitive data and ensure user privacy.

While generative AI promises higher productivity, it relies heavily on humans for guidance. Since GPT can get incorrect answers from the data it consumes, its learning style, and its decision-making capabilities, humans must stay involved throughout the process of AI's creations. Given these limitations, developers may need to implement mechanisms for obtaining user consent, providing transparency and accountability, and enabling users to exercise their rights under data protection laws. In line with the above challenge, this paper introduces a human-in-the-loop methodology for AI-driven refactoring, employing a pipeline for identifying and correcting data clumps in Git repositories.

Some related work pertaining to this section, namely AI-driven software engineering and refactoring, include the following. Refactoring programs using LLMs with few-shot examples is discussed in [23,34]. Classification of factors with AI tools and open problems of the usage were elaborated by [35], such as testing AI systems and overcoming mistrust. A framework and catalogue for prompt engineering technique, showcasing reusable solutions, was elaborated by [31]. In [32] a narrowed focus specifically to the use of LLMs to software engineering tasks was examined. It explores several prompt patterns that have been applied specifically to enhance software engineering processes like requirements' elicitation, rapid prototyping, code quality, deployment, and testing. A thorough systematic literature review explores the applications, impacts, and limitations of LLMs in software engineering by analysing 229 studies, aiming to fill existing knowledge gaps and outline potential areas for future research [36].

### 2.4. Data Clumps and Code Smells in Software Refactoring

Data clumps are recognized as a type of code smell in source code projects [7]. Fowler initially defined data clumps as groups of variables that frequently appear together in various locations within a project, a description that is notably broad. Subsequent research has defined challenges in processing data clumps [37] and has refined this definition to make it more suitable for automatic refactoring purposes [38–40]. Although subjective interpretations of data clumps remain valid, this paper adopts the revised definition to facilitate a discussion on automatic refactoring. According to the literature, data clumps can be classified into three distinct types [39]:

- Field–Field: the data clumps is found between fields or classes.
- Parameter–Parameter: the data clumps is found between parameters of two different methods.
- Parameter–Field: This type emerges during an unfinished refactoring of a parameter–parameter data clumps.

A field–field data clumps is present if specific criteria are met. These criteria help in identifying field–field data clumps by highlighting issues of redundancy, lack of encapsulation, and poor structural organization, suggesting that a refactor may be necessary. An instance for a field–field data clumps can be seen in Figure 1, which shows two classes,

namely *Patient* and *Doctor*. These two classes have various fields. The fields *firstname*, *lastname* and *age* are in common between those classes, showing a redundancy which could be refactored. The criteria for a field–field data clumps are:

- Three or more data fields are shared in two or more classes.
- The data fields have the same signatures, consisting of names, types and visibility.
- Instance fields are not necessarily found in the same order and can be distributed over an instance.

In the scenario of a parameter–parameter data clumps, analogous conditions are considered as those for a field–field data clumps. This situation is exemplified in Figure 2, showcasing two distinct classes: *Patient* and *Doctor*. The methods within these classes exhibit shared parameters, identified as *diagnosis*, *urgency*, and *treatment*. For the sake of simplicity, the types and access levels of these parameters are not specified in this illustration. The criteria for a parameter–parameter data clumps are as follows:

- Three or more input parameters are shared in two or more method declarations.
- The input parameters have the same signatures, consisting of names, types and visibility.
- Method parameters are not necessarily found in the same order.
- The same inheritance hierarchy and method signature should not be present in these methods.

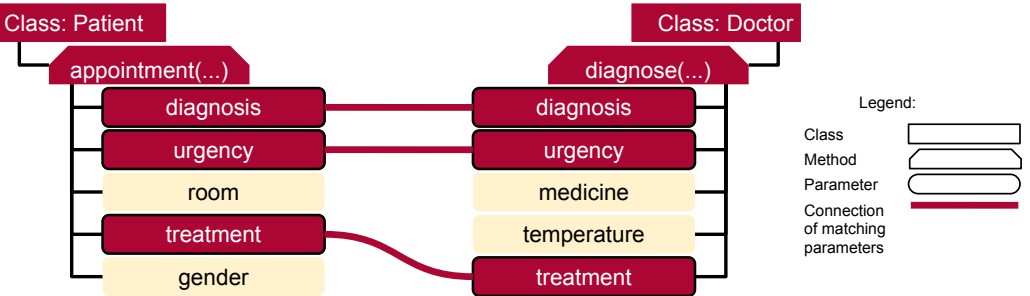

**Figure 1.** Example of a parameter-parameter data clump.

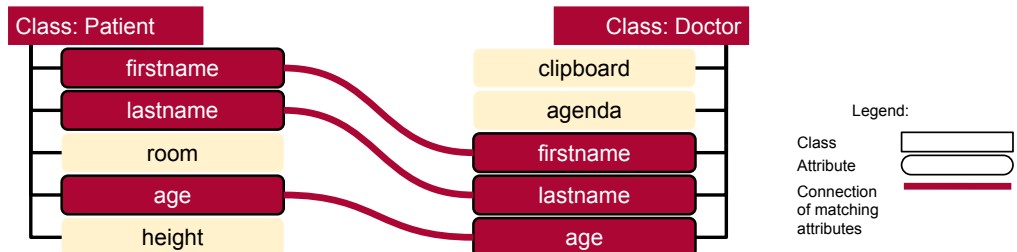

**Figure 2.** Example of a field–field data clump.

In order to refactor data clumps, Fowler already provided a step-by-step solution. Extract fields of a class with data clumps using the *Extract Class* technique and move the fields to their own class. data clumps in parameters shall be refactored using the *Introduce Parameter Object* to extract them off a class. Using the *Preserve Whole Object* technique reduces the size of a parameter list by passing the whole object to a method instead of only the necessary parameters. Fowler also addresses the challenge that after the refactoring new code smells emerge, like a data class, but which can be further enriched by other methods helping to avoid many code duplications.

In 2015, a platform to implement agents for automatic refactoring was proposed [41]. According to the extensive [42] review there existed no tool for data clumps refactoring up to 2020. In 2023 a deep learning-based approach within an IDE plugin was developed using

human feedback [43], but data clumps detection was not implemented. A first approach for a live detection with semi-automatic refactoring of data clumps was then released in 2023 [9].

*2.5. Gap Analysis*

Despite advanced explorations through AI, there still remain numerous gaps and challenges that warrant further research. This is particularly true in the areas of AI-driven refactoring and automated detection and refactoring of data clumps. These issues include the following:

- Limited Integration of AI in data clumps Refactoring: While there is growing interest in leveraging AI for data clumps refactoring, the integration of AI techniques in this domain remains limited. This gap highlights the need for comprehensive exploration and experimentation to unlock the full potential of AI-driven refactoring in addressing data clumps efficiently.
- Challenges of AI-Driven Refactoring: The inherent complexities associated with LLMs pose significant challenges in AI-driven refactoring. These challenges include the risk of generating faulty refactorings and the reliability of AI-generated code, alongside compliance with regulatory and legal requirements. While these challenges are inherently project specific, adopting a human-in-the-loop approach can help mitigate risks and ensure compliance with regulatory standards.
- Lack of Empirical Data for Automated data clumps Refactoring: One of the critical gaps in automated data clumps refactoring is the scarcity of empirical data for verifying functionality. This encompasses both datasets related to detections and feedback from developers regarding refactoring suggestions. Addressing this gap requires the generation of comprehensive datasets, including acceptance feedback from developers, to facilitate the evaluation and improvement of automated data clumps refactoring algorithms.

In addressing some of the identified gaps, this paper introduces an AI-driven tailored pipeline designed specifically for data clump refactoring. Leveraging the capabilities of advanced LLMs like ChatGPT, the pipeline automates the detection and resolution of data clumps, thereby significantly enhancing code quality and maintainability. Furthermore, the integration of a human-in-the-loop methodology enhances the pipeline's effectiveness and trustworthiness. By incorporating human decisions to refine rejected refactorings in subsequent iterations, the pipeline ensures a comprehensive approach to addressing complex coding scenarios. Additionally, this paper underscores the importance of regulatory compliance, particularly in light of standards such as the EU-AI Act. By adhering to the latest regulatory requirements and ethical standards for AI usage in software development, the proposed pipeline ensures responsible and ethical deployment. Further, preliminary experimentation and validation of the pipeline demonstrate promising results in identifying and refactoring data clumps.

## 3. Methodology

This section delineates the architecture of our pipeline, specifically engineered for supporting AI-driven refactoring, with a focus on resolving data clumps. Our methodological approach is founded on a structured pipeline designed to streamline the refactoring process, enhance code maintainability, and facilitate scalability through systematic organization.

The flow diagram Figure 3 outlines our methodological approach, starting with problem identification and culminating in the analysis and conclusions. It includes steps for setting objectives, ensuring compliance with ethical standards, reviewing related work which was conducted in the previous sections in detail. The designing of the AI-driven refactoring pipeline, the integration of AI technologies, and the experimental setup are crucial steps, followed by an execution phase that tests the pipeline's effectiveness. The final analysis assesses the impact of AI on refactoring data clumps, guiding future research directions.

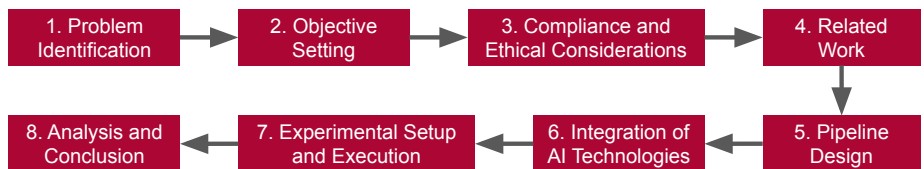

**Figure 3.** Methodology steps for integrating AI in software engineering to refactor data clumps-flowchart diagram.

The use of an AI-driven refactoring approach is applicable when correctly used in the specific field. The integration of AI and ML into the CI/CD process represents a further step into the future, also known as DevOps (Software Development and IT Operations). This allows the cycles of build, test, and deploy to be optimized by performing auto-corrections. Through the use of feedback loops and the analysis thereof, problems can be solved proactively rather than reactively [44].

The pipeline proposed in this work is constructed following a service-based architectural model, in which discrete services are systematically registered to execute distinct tasks. Figure 4 depicts the pipeline, which will be explained in the following. The modular design promotes task reuse across different stages of the pipeline, fostering scalability, maintainability, and flexibility, akin to the principles found in microservices architectures. The architecture is partitioned into three primary components: Handlers, Services, and Context, each playing a pivotal role in the pipeline's functionality:

- Handlers: These components act as intermediaries between the core pipeline logic and the services executing the tasks. Handlers are responsible for managing one or several steps of the pipeline, orchestrating the data flow to and from the services they employ. Their role is crucial in ensuring that the pipeline's processes are executed efficiently and accurately.

- Services: These are the workhorses of the pipeline, tasked with carrying out the computational work required for each step. Designed to be stateless and reusable, they accept specific sets of input data, process these data according to predefined algorithms or tasks, and return the results. This design principle ensures high levels of efficiency and facilitates the easy replacement or updating of individual services without disrupting the pipeline's overall functionality.

- Context: This component acts as a dynamic repository, holding data payloads such as results from previous computations or input data for the next service. The context serves as the communication medium, transferring data from one service to another and ensuring that each service is provided with the necessary information to perform its tasks effectively.

The communication process between the main program, handlers, and services is typically structured as follows and also as shown in Figure 4.

- 1. Initialization: The process begins with the main program initiating the pipeline, establishing the necessary handlers along with their initial context. This stage sets the groundwork for the pipeline's operation, ensuring that all components are correctly aligned and ready to perform their respective tasks.

- 2. Execution: At this stage, for each step of the pipeline, the pipeline step activates the corresponding handler, providing it with the current context received from the preceding step. This ensures a seamless flow of information and maintains the continuity of the pipeline's processes.

- 3. Service Invocation: Upon receiving the context, each handler invokes the appropriate services, forwarding the context enriched with the necessary data extracted from previous operations. This step is pivotal in maintaining the logical flow and ensuring that each service has access to the requisite information.

- 4. Data Processing: The services, each with a distinct objective, complete their designated tasks and relay the outcomes back to the handler. For instance, a service might utilize algorithmic strategies to generate a name for a new class, drawing upon field names provided within the context.

- 5. Progression: Once all required services have been engaged and their tasks completed, the handler compiles the responses and returns the cumulative result back to the main program. The main program then advances to the next step in the pipeline, repeating this process until all predefined steps are concluded.

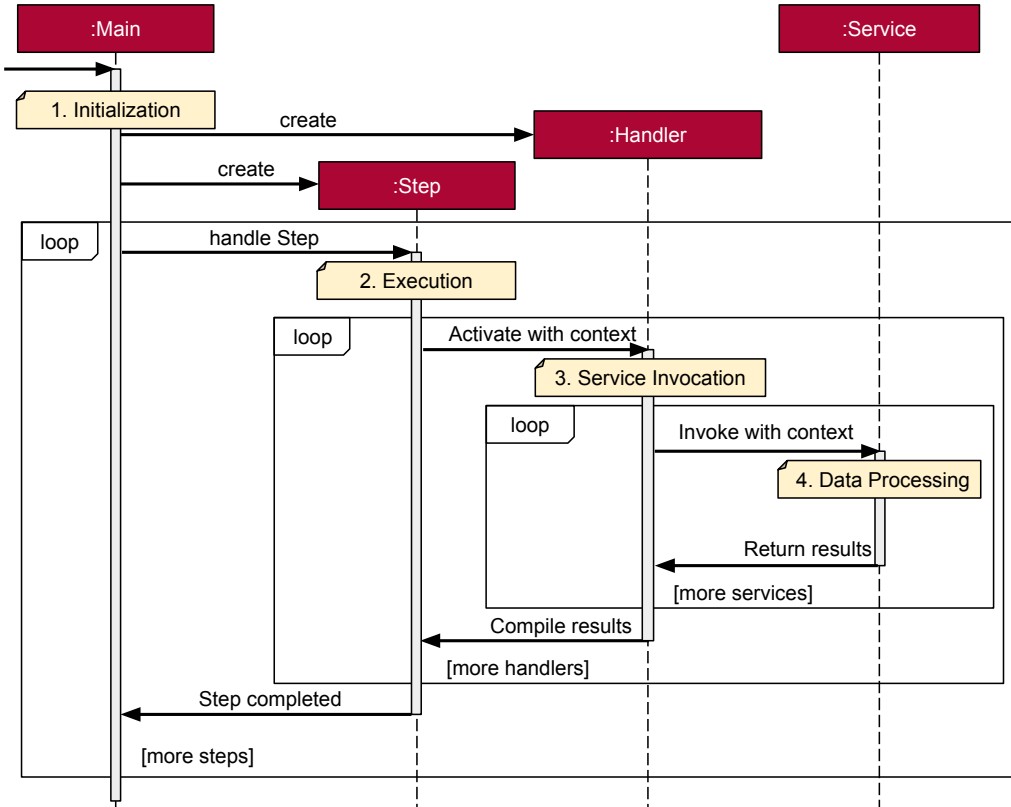

**Figure 4.** Communication process of the pipeline-UML sequence diagram.

Having established the foundational principles and methodologies of our data clumps detection strategy, we now transition to the next crucial phase: the integration of AI technologies into our refactoring pipeline. This advancement is vital for enhancing the automation, efficiency, and effectiveness of our processes, propelling us towards more refined and automated refactoring outcomes.

## 4. Implementation

This section provides a detailed overview of the implementation of our AI-driven refactoring pipeline, specifically designed to address the challenge of data clumps in software projects. We describe the systematic approach adopted in our pipeline, delineating each step from the initial acquisition of source code to the final validation phase. Furthermore, we elaborate on the integration of Large Language Models to enhance the refactoring process, ensuring compliance with recent EU regulations and optimizing the efficiency of our methodologies.

### 4.1. Pipeline Steps

We will now describe our pipeline steps in more detail. Figure 5 illustrates the pipeline steps for our AI-driven data clumps refactoring. The pipeline consists of six steps: (1) Project Obtaining, (2) Data Clumps Detection, (3) Data Clump Prioritization,

(4) Collection of Additional Previous Input, (5) Data Clump Refactoring, (6) Validation and Pull Request Creation.

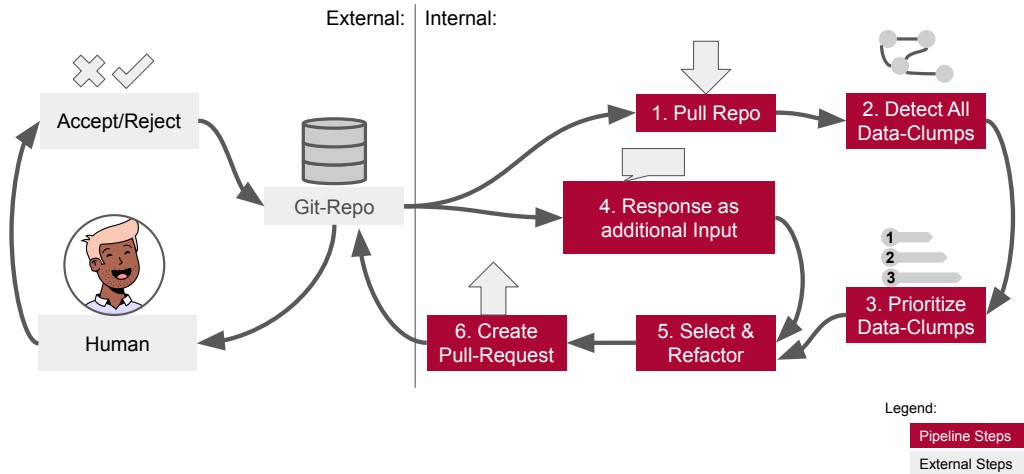

**Figure 5.** Pipeline steps for AI-driven data clump refactoring.

1.  The first step is crucial for obtaining the project to be analyzed. In our illustration, the project is pulled from a Git repository, but this step can be adapted to obtain the project from various sources like GitLab and Subversion.
2.  The second step begins once the source code is fully obtained. In this step, data clumps are detected. The project is first prepared; for instance, unrelated files will be ignored or removed. Depending on the automation mode of the Large Language Model usage, the files will be parsed. If the LLM is used only to generate new class names, the source code files will be parsed into an Abstract Syntax Tree (AST). If the LLM is also handling the refactoring, the source code files will not be further adapted. The choice of automation mode may depend on the project size. Given the nature of data clumps, they may span across the entire software project and create numerous dependencies, resulting in exhaustive analysis. However, this exhaustive analysis faces significant challenges due to the constrained context window of modern LLMs, such as the recently unveiled GPT-4, which is limited to a context window of 128,000 tokens, or approximately 128,000 words. Large software projects may exceed the context window of the LLM, thus reducing the size of the input may be necessary. A possible solution is to either change the LLM or to transform the source code into an Abstract Syntax Tree (AST). The detection of data clumps based on an AST can be performed with an existing tool.
3.  The third step is to prioritize data clumps for refactoring, which is likely to be effective. Currently, there is no prioritization method for data clumps; we omitted this step in our implementation. Prioritizing certain data clumps for refactoring first can have a small but significant impact on reducing faults [45]. Consequently, a more systematic strategy is advocated, wherein identified data clumps are prioritized based on a set of predetermined criteria [46].
4.  Closely connected to the previous step, the fourth step involves gathering additional input such as responses from previous refactorings and responses from the developers of the software project being analysed. In this step, information about the context of the project, its structure, and history can be collected to provide helpful additional input.
5.  This step represents the core of the pipeline and deals with the selection and refactoring of data clumps and is explained in detail in Section 4.2. This step takes into account various factors such as last access time, prioritization, the number of parameters involved, the semantic importance of names, and the specific context of the project, which are results of the previous steps. This stage also introduces the innovative application of AI for refactoring. Depending on the selected automation

mode, different strategies are used. The required files will be presented to the LLM to perform the refactoring or to support the refactoring by providing suitable names for new classes. An additional use of the LLM is to determine the optimal location within the project for the newly established class, ensuring that the refactoring integrates seamlessly into the existing project framework. When the refactoring is performed within a Java project, a handler can use an adapted IntelliJ plugin, ensuring a systematic, deterministic, and secure refactoring process while simultaneously updating the references to the methods, fields, and parameters affected by the changes.

6. The final step of the pipeline handles the validation process to ensure that no errors have been introduced during the automated refactoring and to create a pull request. The validation can be accomplished through one of two primary methods: by employing a recognized build tool such as Gradle, which is capable of orchestrating and executing test cases, or by leveraging CI/CD practices within Git repositories, assuming such practices are already established by the project owners. This ensures a comprehensive and robust verification process, affirming the integrity and effectiveness of the refactoring efforts. By creating a pull request, additional CI/CD procedures may be initiated, which can help further ensure quality control. At this step, a human should be involved to accept or reject the created pull request. This feedback about the acceptance will then be looped into step four as additional input. This pipeline could then be restarted on a new event, such as whenever a new push or merge occurs on a specific branch.

### 4.2. Integration of AI Technologies

The integration of LLMs, such as ChatGPT, into our refactoring pipeline signifies a notable advancement, particularly in the stages of naming and refactoring. The strategy encompasses sophisticated prompt engineering with the LLM, meticulously tailored to efficiently detect and refactor data clumps within our designated research domain. In alignment with the EU regulations on AI transparency, during the final phase of our pipeline, project owners of the refactored repository are informed that the refactoring has been assisted by AI technologies.

We have investigated three primary methodologies for the incorporation of LLMs into our pipeline, evaluated to optimize the balance between active usage and idle time for the LLM, thereby enhancing resource allocation:

- Data Clumps Detection: In this configuration, the LLM is instructed to identify data clumps based on a predetermined definition, outputting the results in JSON format. The actual refactoring process is then executed by an alternative specialized tool.
- Refactoring by the LLM: In this scenario, the LLM is tasked with the refactoring process after the identification of data clumps by a different tool.
- Combined Detection and Refactoring: This innovative approach assigns the LLM the dual responsibilities of identifying and refactoring data clumps, aiming for a seamless integration within the overall pipeline.

The integration of LLMs can be effectively applied during the refactoring process, particularly when generating new class names. Figure 6 demonstrates the refactoring of data clump between two existing classes, namely *Patient* and *Doctor*. In this scenario, the LLM utilizes the attribute fields common to both classes, which include *firstname*, *lastname*, and *age*, to suggest a suitable name for a new class. In the example, the LLM proposes the name *Human* for the new class. While providing the LLM with additional context about the software project may enhance the results, this is not depicted in our selected example.

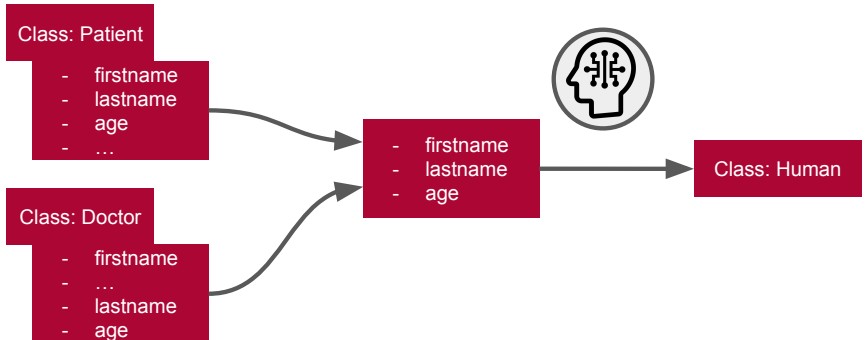

**Figure 6.** Refactoring data clumps with semantic name generation using variables via LLMs.

Support for two LLM platforms has been implemented. The first platform is ChatGPT, as noted in the introduction. The second supported platform is the Ollama platform (https://ollama.com/, accessed on 18 March 2024), which facilitates hosting a LLM without significant resource demands, thereby reducing potential costs. This platform permits users to install multiple LLMs and access them via an HTTP interface. Supported models include Code Llama (https://github.com/facebookresearch/codellama, accessed on 18 March 2024), specifically trained for coding applications. In the works of [47] the potential of using LLMs like GPT-3 was investigated for automatic support of software modelling. They recognized a positive potential for the application of LLMs in this context, underscoring their ability to generate code, predict model elements, and facilitate the evolution of software models.

## 5. Experiments and Results

Although the primary focus of this work is on the architecture of the pipeline for AI-driven data clumps detection, we have extended our experiments to conduct a preliminary study on how various settings and inputs influence the performance of LLMs like ChatGPT in our data clump refactoring tests. Section 5.1 describes our experimental setup, detailing the scenarios. Section 5.2 describes our parameters used for the evaluations. To evaluate the integration of LLMs, we conducted two experiments. The first experiment is based on a small hand created project with around 90 lines of code, while the second experiment uses a real world project with around 180,000 lines of code. The first experiment is described in Section 5.3 and the second in Section 5.4.

### 5.1. Experimental Setup

We restricted the parameter space as follows: We only use GPT-4 as it has shown to produce far-better results than GPT-3.5 We also only transmit all (filtered) files of the project at once in contrast to our approach to submit pairs of files. As the number of files to be transmitted in the latter approach is too enormous, the potential benefits are not justified to perform the experiments.

Please note that, due to a major bug (https://github.com/ollama/ollama/issues/1863, accessed on 3 March 2024) for the Ollama platform, we were unable to run experiments. Our experiments, therefore, were only performed to the OpenAI Apis The following parameters were adjusted for each test scenario. Each test was repeated three times for each combination of parameters:

**Model:** A fundamental choice for our experiments is the model. On the one hand, GPT-4-Turbo provides a larger context size which is essential for refactoring larger projects.

**Instructions:** The instruction format describes how the term *data clumps* is conveyed to the LLM, which is crucial for the success of the model's refactoring or detection of data clumps. We experimented with a *no-definition approach*, an *example-based approach*, and a *definition-based* approach. Each of these settings can show how us how different levels of guidance impact the LLMs performance on detection and refactoring of

data clumps. While the definition-based mirrors our definition from Section 2.4, the example-based approach includes multiple examples of data clumps and how to refactor them. Each example was accompanied by source code comments explaining the data clumps, and a refactored version without any data clumps.

**Data format:** The data format parameter determines how the source code is submitted to the LLM, and therefore strongly impacts the ability of the LLM to understand the source code. When submitting the raw source code, the LLM has a comprehensive view allowing it to detect and refactor data clumps. On the other hand, submitting an AST without method bodies limits its use to detection task as refactoring is difficult to facilitate. Unnecessary information like comments are not included and the AST allows a more structured and unified data format to handle.

**Data size:** Considering that LLMs are often stateless and do not memorize previous interactions, the amount of data sent before waiting for a response is critical. We compared the effect of sending *all project files* at once against transmitting *pairs of files* to assess whether the limited context improves the performance of the LLM to detect data clumps.

**Temperature:** The temperature parameter determines unpredictability of the response by the LLM. The value of the temperature parameter can vary between 0 and 1. A higher temperature leads to more variety in the responses of the LLM, while a lower temperature leads to more stable outputs. By varying this parameter we can assess how a lower or higher temperature affects the detection and refactoring task which helps us find an optimal balance between creativity and accuracy. We choose three temperatures (0.1, 0.5, 0.9) to evaluate whether a low, middle or high value is more suitable for our purpose to detect and refactor data clumps.

### 5.2. Parameters for Evaluation

The experiments were evaluated by comparing the detection and refactoring results of the LLM to the ground truth. A ground truth is the accurate data used as a standard to check the correctness of experimental results. For instance, it is similar to the correct answer key used to grade a test, helping evaluate how well the experiment did. This ground truth was obtained manually and contains all data clumps with their precise location, therefore serving as our benchmark for the accuracy assessment. For all combinations of parameters, the specificity and sensitivity of the detected data clumps were calculated. This approach helps us to evaluate the effect of the different parameters on the accuracy of the LLM, as a data clumps might not be detected or structures might be detected as data clumps even if they are not.

If ChatGPT is asked to return actual source code, a score metric is used to account for the higher flexibility and randomness of source code. In this source metric, correctly refactored source code parts are scored positively while any required refactoring that was not performed or unnecessary refactoring that does not relate to data clumps receives no points. If the returned source code is non-compilable, no points are given. This decision discourages the generation of non-compilable code by favouring a conservative non-action approach over invalid or inaccurate refactoring. While it is possible to fix non-compilable code, we believe that maintaining the code usability is more important than detrimental modifications that could harm the integrity of the codebase.

Specificity (also known as "true negative rate") measures the proportion of actual negatives that are correctly identified by the model. In the context of the work presented in this paper, it represents how well the model identifies parts of the source code that do not contain data clumps. A high specificity means that the model is good at recognizing sections of code that are free of data clumps. In other words, it produced a lot of true negatives.

Another parameter, namely, sensitivity (also known as "recall" or "true positive rate") measures the proportion of actual positives that are correctly identified by the model. In this context, it represents how well the model identifies data clumps that are actually present in the source code. A high sensitivity means that the model is good at detecting data clumps when they exist. In other words, it produced a lot of true positives.

### 5.3. Results of First Experiment

In the first experiment a small test project was created which was analysed by our created pipeline. The project has about 90 lines of code and consists of one parameter–parameter data clump that relates to a Euclidean coordinate, and one field–field data clump that describes a floating point number with an exponent, a mantissa, and a sign. The fields and methods of these data clumps were referenced in multiple parts of the project so that we can validate whether the refactoring would make all necessary changes.

This project is comparably small; we regard the experiment as a preliminary investigation into the efficiency of LLMs in data clump detection and refactoring, which we intend to extend to larger projects in the future. With this approach investigation of smaller projects we can test concentrate to test combination of parameters of the LLMs that have an influence on the quality and usability of these models for detection and refactoring purposes. We believe that such experimentation with parameters is beneficial to reduce the parameter space for further experiments on large-scale projects.

In the following, the results of the first experiment for the detection of data clumps are presented. Both GPT-4 Turbo and GPT-3.5 were able to successfully identify classes with and without data clumps. Section 5.3.1 presents our initial results regarding the detection of data clumps. Section 5.3.2 then discusses initial findings concerning the automated refactoring using different instructional approaches. Finally, Section 5.3.3 explores a combined approach for detecting and refactoring data clumps in a single step.

#### 5.3.1. Detection

However, the median sensitivity of GPT-3.5-Turbo is 0 which indicates that many data clumps were not found by this model. The median specificity is also 0, indicating that GPT-3.5-Turbo detects many false-positives. These are mostly related to data clumps. For instance, information about where a method with a data clump is called, is returned by ChatGPT. While this information is useful for the succeeding steps of the pipeline, it violates our service-based approach by which ChatGPT should only return the information requested, so that other tools can process it correctly.

We also noticed a trade-off between the sensitivity and specificity parameter. If all files were submitted at once, the median sensitivity is at 50%, but the specificity is 14%. This higher sensitivity could be caused by model knowing all information of the source at once and thereby finding more data clumps. However, it might also cause ChatGPT to return more false positives.

A similar trade-off exists for the temperature, as illustrated in Figure 7. Higher temperatures results in a lower sensitivity, while higher temperatures are correlated with a higher specificity. The positive correlation is noteworthy, as we assumed that higher temperatures would lead the LLM to detect more unrelated pattern as data clumps. However, with higher temperature the model deviates more from its training, which might be helpful for data clump detection.

The choice of using the source code or the AST tends to have less impact on our result. However, submitting the AST has a better median specificity and sensitivity than the source code. One reason for this could be that omitting the method bodies helps the model to focus more on the relevant information for data clump detection.

The impact of the instruction format is also visible. Providing ChatGPT with example is the best approach with a median sensitivity of 48%. Providing no definition leads to a sensitivity of 35% while providing a rigid definition of data clumps seems to cause less accurate results. This could be caused by our data clump definition, which might not be rigid enough.

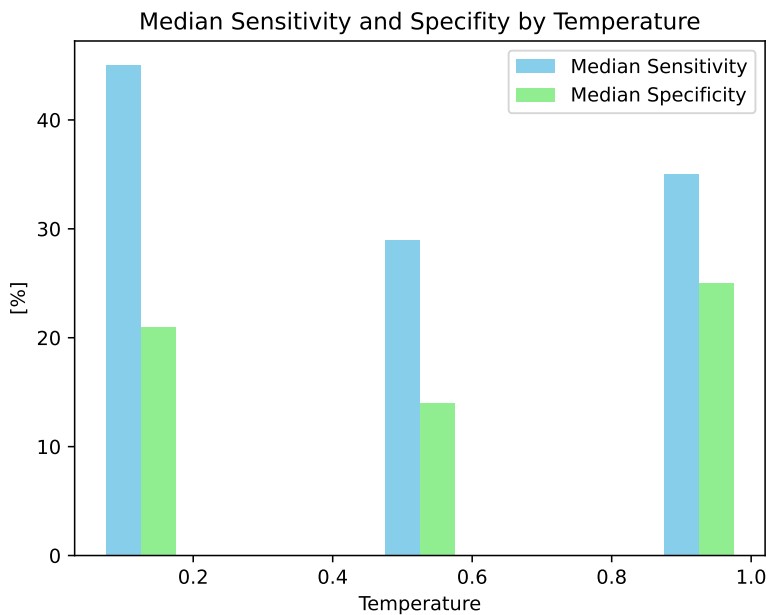

**Figure 7.** Median sensitivity and specificity in relation to the temperature.

### 5.3.2. Refactoring

If ChatGPT is prompted to refactor source code while also given the locations of all data clumps, the median of GPT-4 and GPT-3.5 is identical (68%) The arithmetic mean of GPT-3.5-Turbo is less which could be explained by the existence of more compiler errors. Similarly, the median of the three instruction variants is identical. For the temperature, we observed that higher temperatures result in a median of 0%, which means that the increased creativity of ChatGPT produces more often code that is non-compilable.

### 5.3.3. Detection and Refactoring

If data clumps are to be detected and refactored at the same time by the model, the weaknesses of GPT-3.5-Turbo become visible. The median score of GPT-3.5-Turbo is 7% in comparison to 82% of GPT-4. The effect of the instruction type is a contrast to our expectation. Providing no definition of data clumps leads to the best results, while the median of the other instruction types are 0% each. Consequently, the mean of the no-definition approach is 46%, while the example-based and definition-based approach are very close to each other (30%).

One reason for this behaviour could be overloading of ChatGPT, since it has to parse the instruction or the examples and apply them to the source code, which might be too much to do in one step.

However, like the refactoring-only approach, we observe that a high temperature leads to worse results, probably caused by compiler errors. This is another indication that low temperatures are more useful for refactoring purposes.

### 5.4. Results of the Second Experiment

The second experiment was conducted after the initial experiment was finished. We used the insights and experiences of first experiment to better determine the parameter space and configuration of the second experiment. In this experiment, we use ArgoUml as a project to analyze. With about 180,000 lines of code, this is project is too large to be reasonably processed. Consequently, we sampled a subset of the data clumps so that the number of data clumps is at least three while minimizing the size of the files that encompass the data clumps. This approach balances the need to limit the size of the transferred files while ensuring that the experiment is more realistic than the first experiment. Higher

threshold with regards to the number of data clumps increased the resource usage strongly so that we chose a more cautious approach. To mitigate the effect of the small data clump count, we conducted 10 experiments per configuration.

The results of the second experiments show that even by filtering and reducing the amount of data, the effect of the large transmission is visible. For instance, if data clumps are defined by giving examples, the amount of data to transfer is comparably larger than giving a simple definition or no definition at all. This is could be one reason why the median sensitivity and specificity is 0%, while the median specificity and sensitivity for the other instruction formats is 100%. The high median of the latter can be explained by the small amount of data clumps and the increased number of tests per configuration.

With regard to temperature parameter, we see that the effect on the median specificity is non-visible as the median is always 100%. Also, the median specificity is worse but unaffected by the temperature. If we consider the mean instead of the median, we see a difference which can be explained by the existence of outliers. A temperature of 0.5 results in a higher mean sensitivity of 78% compared to the temperature 0.1 (62%) and 57% for the temperature 0.9. Our observation from the initial experiment that a high temperature of 0.9 leads to a better sensitivity cannot be confirmed in the second experiment. A temperature of 0.5 results in a specificity of 29%, while the mean specificity of the temperature 0.1 is 23% and 21% for the temperature 0.9. However, since the median is constant and the difference is small, the effect of the temperature is not as strong as expected.

Considering the data format (AST or complete source code), the advantages of transmitting fewer data becomes apparent. While the median sensitivity and specificity of transmitting the complete source code is 0%, the median sensitivity of transmitting the AST is 100%, meaning that data clumps are found when providing the AST but not when only transferring the source code alone. The median specificity of the AST data format is 33%. The mean shows a similar gross contrast as the source-code approach has a median sensitivity of 32% compared to a mean specificity of 92%. A smaller contrast exists between the mean specificity of the source-code approach (10%) and the AST-approach (35%).

We also performed preliminary refactoring experiments on ArgoUML using a LLM, but the results are unsatisfactory. The source code could not compile with some exceptions in the lower tenths range. The refactored code by the LLM must be modified by a human-in-the-loop before it could compile. For instance, during our preliminary tests, methods are not refactored or the extracted class is not generated. We also notice a certain "laziness" of ChatGPT, as it might return only some parts of the original source code, so that the human-in-the-loop must construct a working source code manually. Therefore, the results of ChatGPT cannot be saved directly in a file but must be processed beforehand. A full scale evaluation of the refactoring quality is challenging and could not be performed by us at the time given that the scope of our study is the creation of a pipeline. This indicates that a further configuration of our tested model ChatGPT is required, in order to successfully automatically refactor data clumps.

## 6. Discussion

The discussion of the experimental analysis reveals several key observations and considerations as outlined below:

- Performance and efficacy: The AI-driven refactoring pipeline was designed to analyse and refactor projects affected by data clumps. The performance of the pipeline was contingent on the size of the projects being tested. While the pipeline facilitates the automatic refactoring of data clumps, its efficacy significantly depends on the underlying LLM employed for the refactoring process.
- Validation of results: In general, it is observed based on the conducted experiments that LLMs like ChatGPT can be used for all three approaches. However, validation of the result remains important, as the accuracy is sometimes unsatisfactory. More sophisticated prompt engineering, can be a useful method to improve the results, since it can be seen that the effect of the parameters is noticeable. With this pipeline we

can now delve further in a more structured analysis with more promising results as our current experiments lack realistic project sizes (lines of code and files) and more detailed analysis of the impact of the parameters. Such a quantitative or qualitative analysis can assess the pipeline's impact on code quality. A more thorough comparison between our proposed method and existing state-of-the-art techniques could enhance the significance of our results. This comparison could be grounded on a shared dataset or benchmarks that have broad recognition within the software research domain. Regrettably, we are not aware of any such datasets specifically tailored for data clumps with refactoring results, which complicates the execution of such a comparison.

- Non-compilable code: The effect of non-compilable code was greater than expected. In most cases, manually correcting them is feasible. For instance, constructors are missing or overridden methods are not updated. However, this is still a time-consuming task that defeats our idea of automatic refactoring. Also, some outliers were not valid Java code at all. For instance, ChatGPT sometimes returns a message stating that no data clumps were detected in a specific file. In some other edge cases, new lines terminators "\n" were not escaped properly, leading also to invalid Java code. These edge cases highlight the need to verify the LLM output. While, these errors can be detected, correcting them is very challenging so that further research is needed. Thus, challenges arising from non-compilable code were notable, often necessitating manual intervention to generate valid source code.

- Daily token limit: During our experiments, we frequently reached the daily token limit when using ChatGPT, underscoring the high demand for computational resources. Moreover, the associated costs and computational requirements present considerable challenges, especially for large-scale projects. A primary issue identified was the project size; each additional method or class necessitated a cross-check with all other classes and methods for data clump detection, exacerbating the complexity and resource demands.

- Project size: Our initial results suggest that ChatGPT is adept at refactoring smaller projects currently on its alone. However, the impact of various parameters is significant and warrants careful consideration before employing a LLM for refactoring tasks. Although general trends were discernible in our sample project, these results should not be deemed representative of all cases. A thorough preliminary analysis is recommended to ascertain the optimal configuration of LLMs for superior results. Overall, it can be stated that the primary challenge remains w.r.t to the management of large-scale projects, which is currently impractical due to computational and cost constraints. A potential solution involves segmenting the project into smaller, more manageable parts. While this approach facilitates more granular analysis, it could result in an increased number of required prompts, thereby indirectly inflating costs and rendering the widespread implementation of this strategy unfeasible. Additionally, software projects that have no clear identifiable components or modules make it difficult to determine how the segments should be chosen. While our primary objective is to develop a pipeline capable of accurately and reliably refactoring data clumps, further exploration into enhancing its scalability is desired.

- Additional parameters: Including more parameters could influence our results. For instance, follow-up questions are a popular tactic to improve the quality of LLM by instructing the LLM to find more data clumps with or without modifying the information available to the LLM. Additionally, the wording of our instruction, or the exact structure of our data clumps examples might be considered as a parameter worth varying. However, the work presented here encompasses initial results based on preliminary experimentation. Thus, considering the associated costs and resource requirements, testing multiple variants is an item for future work.

- Comprehensive comparison: Our current work focuses on presenting the core structure and approach of our AI-driven pipeline for data clump refactoring. While the results demonstrate the functionality and potential of our method, a more in-depth

evaluation would provide valuable insights for further development. A more detailed performance analysis and comprehensive method comparison would be beneficial, specifcally on how to deal with the limitations of AI-driven refactoring.

In summary, while AI-driven refactoring shows promise, particularly for smaller projects, addressing challenges related to project size, non-compilable code, and resource constraints is essential for broader applicability. Further research and refinement of methodologies are necessary to overcome these obstacles and unlock the full potential of AI-driven refactoring in software engineering.

## 7. Conclusions

The application of AI in refactoring data clumps is promising, despite encountering challenges. Currently, leveraging LLMs for real-world projects, which often contain thousands of classes and are of substantial size, seems impractical. A more viable approach at this juncture appears to be the integration of deterministic algorithms assisted by LLMs to generate context-specific information, such as the naming of new classes. Furthermore, with the introduction of new AI regulations by the EU, refactoring using AI presents additional challenges that necessitate a robust and secure methodology, accompanied by human oversight. Additionally, the implementation of a well-structured CI/CD pipeline with verification processes is essential, as it aids in validating the correctness of the refactoring by executing tests.

Acknowledging current limitations, we opt for a phased approach, incorporating AI on a smaller scale within pipeline steps to manage and control costs, reduce time consumption, and minimize potential risks. This approach allows for a more manageable and gradual integration of AI capabilities into our processes. Our findings suggest that the utilization of AI for refactoring data clumps is currently not efficient for processing entire files or projects at once. Future research will explore the integration of deterministic algorithms for refactoring, coupled with AI-assisted naming of new files and contextual identification for improved outcomes.

For future work, we plan to continue testing the refactoring of data clumps with newer LLMs. We also aim to adapt our modular pipeline for integration with an IntelliJ plugin, enhancing it with AI for context-specific tasks such as naming new classes. This will be tested with actual repositories to obtain more detailed insights into the accuracy of the refactoring process.

**Author Contributions:** Conceptualization, N.B.; methodology, N.B. and T.S.; software, T.S.; validation, N.B., P.I. and T.S.; formal analysis, T.S.; investigation, N.B., P.I. and T.S.; resources, N.B.; data curation, N.B. and T.S.; writing—original draft preparation, N.B.; writing—review and editing, N.B., P.I. and T.S.; visualization, N.B.; supervision, P.I. and E.P.; project administration, N.B. and P.I.; funding acquisition, T.S. All authors have read and agreed to the published version of the manuscript.

**Funding:** This research was supported by the provision of computational credits valued at USD 5000 from the OpenAI Researcher Access Program, intended for ChatGPT API usage. A significant portion of these credits remains unused as we initially focused on developing our pipeline before proceeding with detailed testing.

**Data Availability Statement:** All the data presented are available within this article. The data set for the experiments can be found on GitHub https://github.com/compf/data_clump_solver, accessed on 18 March 2024.

**Conflicts of Interest:** Author Padma Iyenghar was employed by the company Innotec GmbH. The remaining authors declare that the research was conducted in the absence of any commercial or financial relationships that could be construed as a potential conflict of interest.

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
