# Peer review of "AI-Driven Refactoring: A Pipeline for Identifying and Correcting Data Clumps in Git Repositories"

_electronics, doi:10.3390/electronics13091644_

Round 1
Reviewer 1 Report
Comments and Suggestions for Authors
This paper proposes to an LLM-based pipeline to identify the potential data clumps of git repositories and refactor them. I think the idea presented in the paper is very interesting, although the novelty and the technical aspects are not that strong. Even though the proposed method mostly relies on the existing LLM (e.g., GPT), the authors pose a potentially more feasible way to resolve data clumps, which look common and trivial, but actually are very tedious and time costly to be resolved in practical software development. My detailed comments are as follows:
1. In Section 2.2, the authors describe some limitations of the present LLM, but more importantly, how do these limitations affect an AI-driven software development? In specific for this work, how do the authors consider the existence of these limitations for the proposed refactoring method?
2. Similarly, in Section 2.3, the authors list some critical challenges in the usage of AI-driven refactoring. How do the authors consider these challenges for the proposed method? How can the authors quantify how well the proposed method addresses these challenges?
3. In Section 5.1, the authors validate the method by using a project with 90 lines, which seems a very simple example. I would suggest the authors repeat the experiment on more complicated code and evaluate the performance. More importantly, by repeating the experiment on a complicated project, the scalability of the method can be evaluated, so it will be possible to see whether this pipeline can be used for more a practical software project.
4. In Section 5, a quantitative compression between the proposed method and state-of-the-art techniques, described in Section 2.4, is missing.
Author Response
Dear reviewer,
We sincerely appreciate your insightful comments and valuable suggestions regarding our work on developing a pipeline using Large Language Models (LLMs) to identify and refactor data clumps. In response, we have carefully considered each of your points and have detailed below how we have integrated your feedback into our work. In instances where incorporating a suggestion was not feasible, we have provided a comprehensive explanation to clarify our rationale. Thank you for your invaluable contribution to refining our research.
- Limitations and AI-Driven Software Development:
“In Section 2.2, the authors describe some limitations of the present LLM, but more importantly, how do these limitations affect an AI-driven software development? In specific for this work, how do the authors consider the existence of these limitations for the proposed refactoring method?”
We appreciate the reviewer's suggestion for a more comprehensive overview of limitations in Section 2.2. We acknowledge that certain limitations, such as the need for human oversight, remain unsolved in current LLM technology. Our proposed pipeline addresses this by strategically outsourcing these limitations to the user.
To further enhance the discussion, we included at the end of Section 2.2 a paragraph about the effect of these limitations. We still consider a human in the loop as necessary.
- Quantifying Challenges and LLM Performance:
“Similarly, in Section 2.3, the authors list some critical challenges in the usage of AI-driven refactoring. How do the authors consider these challenges for the proposed method? How can the authors quantify how well the proposed method addresses these challenges?”
We concur with the reviewer's concern regarding quantification of the proposed method's effectiveness in addressing AI-driven refactoring challenges. As another reviewer raised a similar point, we agree that further analysis is necessary to definitively measure LLM performance on data clump refactoring.
- In-depth user acceptance analysis: We integrated a mechanism within the pipeline to track user acceptance of refactoring suggestions by Git repository maintainers. This data will provide valuable insights into how effectively the LLM tackles real-world data clump refactoring scenarios.
- Benchmarking on a larger dataset: We are actively exploring the possibility of utilizing established software engineering benchmarks or creating a dedicated data clump refactoring dataset, whose existence at the time is not known to us. Such a dataset could be used for a more quantitative comparison of our method's performance against other approaches.
We revised Section 6 (Discussion) and included that such in-depth analysis and benchmarking are useful to compare methods in AI-driven refactoring for data clump refactoring.
- Scalability and Experiment Complexity:
“In Section 5.1, the authors validate the method by using a project with 90 lines, which seems a very simple example. I would suggest the authors repeat the experiment on more complicated code and evaluate the performance. More importantly, by repeating the experiment on a complicated project, the scalability of the method can be evaluated, so it will be possible to see whether this pipeline can be used for more a practical software project.”
The reviewer's point regarding the limited project size (90 lines) in Section 5.1 is well-taken. We acknowledge the wish for a deeper evaluation of the method's scalability in handling more complex codebases. To address this, we conducted an additional experiment using ArgoUml with significantly larger codebases (180,000 LOC). This will provide a more comprehensive assessment of the pipeline's ability to effectively identify and refactor data clumps within practical software development scenarios. While further analysis with more and larger codebases is planned, such a comprehensive analysis is out of the scope of our current paper, as this paper focuses on the creation of a pipeline with reliable results. For the planned analysis we are going to create pull requests to open source projects and analyze the maintainers' responses.
- Quantitative Comparison with Existing Techniques:
“ In Section 5, a quantitative compression between the proposed method and state-of-the-art techniques, described in Section 2.4, is missing.”
The reviewer's observation regarding a missing quantitative comparison between our method and existing techniques is accurate. However, Section 2.4 focuses on providing background information on data clumps and refactoring techniques, not a comparative analysis.
Our primary focus lies on improving the reliability of data clump detection and refactoring, not necessarily outperforming existing methods. While a future in-depth comparison could be valuable, such an endeavor hinges on the availability of a widely recognized benchmark dataset specifically designed for data clump refactoring. Unfortunately, no such dataset is currently known to us, making a direct quantitative comparison challenging.
Despite this obstacle, we acknowledge the potential benefit of a future comparison and will include a discussion of this point in a revised Section 6 (Discussion). We will emphasize our commitment to exploring the feasibility of creating or utilizing an appropriate benchmark dataset to facilitate future quantitative comparisons.
Reviewer 2 Report
Comments and Suggestions for Authors
Review Report
Journal: Electronics (ISSN 2079-9292)
Manuscript ID: electronics-2951527
Article Type: Article
Title: AI-Driven Refactoring: A Pipeline for Identifying and Correcting Data Clumps in Git Repositories
Authors: Nils Baumgartner, Padma Iyengar, Timo Schoemaker and Elke Pulvermüller
Summary: This study focuses on identifying and refactoring data clumps within software systems, which are signs of poor code structure and can lead to various challenges. An innovative pipeline driven by advanced AI models, including Large Language Models (LLMs) like ChatGPT, is introduced to automate this process, improving code quality and maintainability. The approach also ensures compliance with regulatory and ethical standards, such as the EU-AI Act, by incorporating human oversight. Initial experiments confirm the effectiveness of the approach, although some challenges in utilizing LLMs are acknowledged.
However, this work in the present form needs a Minor revision due to the following reasons:
1. The authors should elaborate on the AI-driven pipeline for better comprehension and technical insight. They should provide clear descriptions of methodologies, algorithms, and techniques for reproducibility.
2. The authors should compare their approach comprehensively with existing methods to highlight its novelty and significance. They should analyze performance against alternative methods for stronger research validation.
3. The authors should include quantitative or qualitative assessments of the pipeline's impact on code quality. They should incorporate metrics or case studies to substantiate their claims.
4. The authors should enhance the reproducibility and availability of their pipeline. They should provide access to resources like source code and datasets to facilitate further experimentation.
Author Response
Dear reviewer,
We sincerely appreciate your insightful comments and valuable suggestions regarding our work on developing a pipeline using Large Language Models (LLMs) to identify and refactor data clumps. In response, we have carefully considered each of your points and have detailed below how we have integrated your feedback into our work. In instances where incorporating a suggestion was not feasible, we have provided a comprehensive explanation to clarify our rationale. Thank you for your invaluable contribution to refining our research.
- Detailed Descriptions of Methodologies, Algorithms and Reproducibility
“The authors should elaborate on the AI-driven pipeline for better comprehension and technical insight. They should provide clear descriptions of methodologies, algorithms, and techniques for reproducibility.”
We agree that providing more detailed descriptions of methodologies, algorithms, and techniques could be very helpful for readers with a strong technical background.
To promote reproducibility, we'd like to highlight that the complete source code and project are available in the "Data Availability Statement" section, as mandated by ACM. This includes all relevant artifacts necessary to replicate our work.
We acknowledge that a more in-depth exploration of the technical aspects would be beneficial for highly technical readers. We may consider addressing this level of detail in a separate publication focused specifically on the technical implementation of the pipeline.
- Comprehensive Comparison
“The authors should compare their approach comprehensively with existing methods to highlight its novelty and significance. They should analyze performance against alternative methods for stronger research validation.”
We appreciate the reviewer's suggestion for an even more comprehensive comparison with existing methods to strengthen the validation of our approach. While expanding the current paper's scope to include a full-scale comparison might not be ideal for maintaining a focused presentation, we've taken steps to enhance the discussion in Section 6. We've:
- Extended the existing discussion: We emphasize that our current work is missing a full-scale comparison and a more in-depth analysis would be beneficial.
- Highlighted novelty: Clarified that our proposed method in comparison to existing approaches is capable of refactoring data clumps..
For a more in-depth performance analysis and comprehensive method comparison, we believe a dedicated study would be most effective. This future work could involve a broader range of methods and a thorough evaluation using a broader range of performance metrics.
- Further Quantitative or Qualitative Experiments
“The authors should include quantitative or qualitative assessments of the pipeline's impact on code quality. They should incorporate metrics or case studies to substantiate their claims.”
We fully agree with the reviewers opinion. As another reviewer also mentioned this we investigated a larger codebase, namely ArgoUml, with around 180,000 LOC. These results are added at Section 5.1 and 5.2. Further structured analysis is planned and desired by us by investigating public git repositories and collecting the maintainers' responses. This is exactly why we first created a pipeline in order to further analyze the acceptance, size, subjective correctness and other factors in-depth. We added a small passage in the discussion Section 6 which talks about the lack of in depth analysis. Also the usage of well chosen metrics or case studies would be beneficial for our work.
- Reproducibility, Source Code and Datasets
“The authors should enhance the reproducibility and availability of their pipeline. They should provide access to resources like source code and datasets to facilitate further experimentation.”
The access to resources is given at “Data Availability Statement” in the end as required by ACM. This includes all relevant artifacts necessary to replicate our work.
Reviewer 3 Report
Comments and Suggestions for Authors
Dear authors , although this word refers an updated topic , related with AI, my suggestion is to include a flow diagram that represent the methodology steps followed at this work.
Could you verify the number of referencias, especially 33 and 34 references ,it seems there are not cited .
Author Response
Dear reviewer,
We sincerely appreciate your insightful comments and valuable suggestions regarding our work on developing a pipeline using Large Language Models (LLMs) to identify and refactor data clumps. In response, we have carefully considered each of your points and have detailed below how we have integrated your feedback into our work. In instances where incorporating a suggestion was not feasible, we have provided a comprehensive explanation to clarify our rationale. Thank you for your invaluable contribution to refining our research.
- Visual Methodology Representation
“Dear authors , although this word refers an updated topic , related with AI, my suggestion is to include a flow diagram that represent the methodology steps followed at this work.”
We appreciate the reviewer's suggestion for a visual representation of our methodology. To enhance clarity, we have included a flow diagram at the beginning of Section 3 that outlines the key steps involved in our approach.
- Missing Citations for References:
“Could you verify the number of referencias, especially 33 and 34 references ,it seems there are not cited .”
The reviewer found missing citations for references 33 and 34 is accurate. We apologize for this oversight. We have reviewed the manuscript and ensured that both references 33 and 34 are now properly cited within Section 4 (Implementation).